# LABEL SMOOTHED EMBEDDING HYPOTHESIS FOR OUT-OF-DISTRIBUTION DETECTION

## ABSTRACT

Detecting out-of-distribution (OOD) examples is critical in many applications. We propose an unsupervised method to detect OOD samples using a $k$-NN density estimate with respect to a classification model's intermediate activations on in-distribution samples. We leverage a recent insight about label smoothing, which we call the *Label Smoothed Embedding Hypothesis*, and show that one of the implications is that the $k$-NN density estimator performs better as an OOD detection method both theoretically and empirically when the model is trained with label smoothing. Finally, we show that our proposal outperforms many OOD baselines and we also provide new finite-sample high-probability statistical results for $k$-NN density estimation's ability to detect OOD examples.

## 1 INTRODUCTION

Identifying out-of-distribution examples has a wide range of applications in machine learning including fraud detection in credit cards (Awoyemi et al., 2017) and insurance claims (Bhowmik, 2011), fault detection and diagnosis in critical systems (Zhao et al., 2013), segmentations in medical imaging to find abnormalities (Prastawa et al., 2004), network intrusion detection (Zhang & Zulkernine, 2006), patient monitoring and alerting Hauskrecht et al. (2013), counter-terrorism (Skillicorn, 2008) and anti-money laundering (Labib et al., 2020).

Out-of-distribution detection is highly related to the classical line of work in anomaly and outlier detection. Such methods include density-based (Ester et al., 1996), one-class SVM (Schölkopf et al., 2001), and isolation forest (Liu et al., 2008). However, these classical methods often aren't immediately practical on large and possibly high-dimensional modern datasets.

More recently, Hendrycks & Gimpel (2016) proposed a simple baseline for detecting out-of-distribution examples by using a neural network's softmax predictions, which has motivated many works since then that leverage deep learning (Lakshminarayanan et al., 2016; Liang et al., 2017; Lee et al., 2017). However, the majority of the works still ultimately use the neural network's softmax predictions which suffers from the following weakness. The uncertainty in the softmax function cannot distinguish between the following situations: (1) the example is actually in-distribution but there is high uncertainty in its predictions and (2) the example is actually out of distribution. This is largely because the softmax probabilities sum to 1 and thus must assign the probability weights accordingly. This has motivated recent explorations in estimating *conformal sets* for neural networks (Park et al., 2019; Angelopoulos et al., 2020) which can distinguish between the two cases.

In this paper, we circumvent the above-mentioned weakness by avoiding using the softmax probabilities altogether. To this end, we approach OOD detection with an alternative paradigm – we leverage the intermediate embeddings of the neural network and nearest neighbors. Our intuition is backed by recent work in which the effectiveness of using nearest-neighbor based methods on these embeddings have been demonstrated on a range of problems such as uncertainty estimation (Jiang et al., 2018), adversarial robustness (Papernot & McDaniel, 2018), and noisy labels (Bahri et al., 2020).

In this work, we explore using $k$-NN density estimation to detect OOD examples by computing its density on the embedding layers. To this end, it's worth noting that $k$-NN density estimation is a *unsupervised* technique, which makes it very different from the aforementioned deep $k$-NN work (Bahri et al., 2020) which leverages the label information of the nearest neighbors. One key intuition

here is that low $k$-NN density examples might be OOD candidates as it implies that these examples are far from the training examples in the embedding space.

In order for density estimation to be effective on the intermediate embeddings, the data must have good *clusterability* (Ackerman & Ben-David, 2009), meaning that examples in the same class should be close together in distance in the embeddings, while examples not in the same class should be far apart. While much work has been done for the specific problem of clustering deep learning embeddings (Xie et al., 2016a; Hershey et al., 2016) many of these ideas are not applicable to density estimation.

In this paper, we use a much simpler but effective approach of label smoothing, which involves training the neural network on a *soft* label obtained by taking a weighted average between the original one-hot encoded label and the uniform distribution over labels. We leverage a key insight about the effect of label smoothing on the embeddings (Müller et al., 2019): training with label smoothing has the effect of *contracting* the intermediate activations of the examples within the same class to be closer together at a faster rate relative to examples in different classes. This results in embeddings that have better clusterability. We call this the *Label Smoothed Embedding Hypothesis*, which we define below.

**Hypothesis 1** (Label Smoothed Embedding Hypothesis (Müller et al., 2019))**.** *Training with label smoothing contracts the intermediate embeddings of the examples in a neural network, where examples within the same class move closer towards each other in distance at a faster rate than examples in different classes.*

We refer interested readers to Müller et al. (2019) for 2D visualizations of this effect on the model's penultimate layer. We will later portray the same phenomenon using $k$-NN density estimation.

We summarize our contributions as follows:

**(1)** We propose a new procedure that uses label smoothing along with aggregating the $k$-NN density estimator across various intermediate representations to obtain an OOD score.

**(2)** We show a number of new theoretical results for the $k$-NN density estimator in the context of OOD detection, including guarantees on the recall and precision of identifying OOD examples, the preservation of the ranking w.r.t. the true density, and a result that provides intuition for why the Label Smoothed Embedding Hypothesis improves the $k$-NN based OOD score.

**(3)** We experimentally validate the effectiveness of our method and the benefits of label smoothing on benchmark image classification datasets, comparing against recent baselines, including one that uses $k$-NN in a different way, as well as classical alternatives to the $k$-NN but applied in the same way. The comparison against these ablative models highlight the discriminative power of the $k$-NN density estimator for OOD detection.

**(4)** We conduct ablations to study the performance impact of the three hyper-parameters of our method - (1) the amount of label smoothing, (2) which intermediate layers to use, and (3) number of neighbors $k$.

## 2 ALGORITHM

We start by defining the foundational quantity in our method.

**Definition 1.** *Define the $k$-NN radius of $x \in \mathbb{R}^D$ as*

$$r_k(x; X) := \inf\{r > 0 : |X \cap B(x, r)| \geq k\}.$$

*When $X$ is implicit, we drop it from the notation for brevity.*

Our method goes as follows: upon training a classification neural network on a sample $X_{\text{in}}$ from some distribution $f_{\text{in}}$, the intermediate representations of $X_{\text{in}}$ should be close together (in the Euclidean sense), possibly clustered by class label. Meanwhile, out-of-distribution points should be further away from the training manifold - that is, $r_k(g_i(x_{\text{out}}); X_{\text{in}}) > r_k(g_i(x_{\text{in}}); X_{\text{in}})$ for $x_{\text{in}} \sim f_{\text{in}}, x_{\text{out}} \sim f_{\text{out}}$, where $g_i$ maps the input space to the output of the $i$-th layer of the trained model. Thus, for fixed

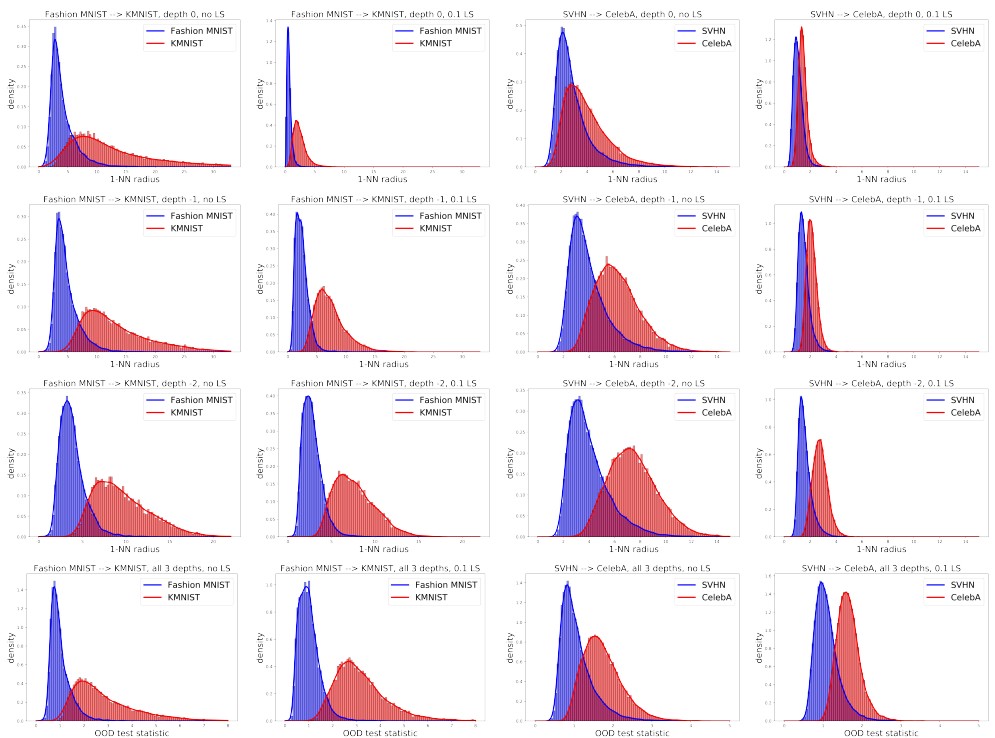

Figure 1: Distributions of the 1-NN radius distance for each of three layers as well as our test statistic, which aggregates over all layers. "Depth" refers to the layer index with respect to the logits layer. Thus, 0 means logits, -1 means the layer right before logits, so on and so forth. The first two (last two) columns correspond to the dataset pairing Fashion MNIST → KMNIST (SVHN → CelebA) with and without label smoothing. The Fashion MNIST pairing uses a 3 layer feedforward neural network while SVHN uses the convolutional LeNet5. Blue represents samples from the test split of the dataset used to train the model and are therefore inliers. Red represents the out-of-distribution samples. Consider the cases of no label smoothing. We see that there is separability between in and out points at each layer, generally more so at deeper (earlier) layers of the network. This motivates our use of $k$-NN distance for OOD detection. There is, however, non-trivial overlap. Now observe the cases with $\alpha = 0.1$ smoothing. The 1-NN radii shrink, indicating a "contraction" towards the training manifold for both in and out-of-distribution points. The contraction is, however, higher for ID points than for OOD points. This motivates the use of label smoothing in our method.

layer $i$, we propose the following statistic:

$$T_i(x) := \frac{r_k\left(g_i(x); g_i(X_{\text{in}})\right)}{Q(X_{\text{in}}, g_i)}, \quad Q(X_{\text{in}}, g_i) := \mathbb{E}_{z \sim f_{\text{in}}} r_k\left(g_i(z); g_i(X_{\text{in}})\right).$$

Since $Q$ depends on unknown $f_{\text{in}}$, we estimate it using cross-validation:

$$\hat{Q}(X_{\text{in}}, g_i) = \frac{1}{|X_{\text{in}}|} \sum_{x \in X_{\text{in}}} r_k(g_i(x); g_i(X_{\text{in}} \setminus \{x\})) = \frac{1}{|X_{\text{in}}|} \sum_{x \in X_{\text{in}}} r_{k+1}(g_i(x); g_i(X_{\text{in}})).$$

Letting $\hat{T}_i$ be our statistic using $\hat{Q}$, we now aggregate across $M$ layers to form our final statistic:

$$\hat{T}(x) = \frac{1}{M} \sum_{i=1}^{M} \hat{T}_i(x).$$

We use a one-sided threshold rule on $\hat{T}$ - namely, if $\hat{T} > t$ we predict out-of-distribution, otherwise we do not. With key quantities now defined, we use the $k$-NN radius to substantiate (1) the claim that in and out-of-distribution points are different distances away from the training points and (2)

Hypothesis 1, that label smoothing causes in-distribution points to contract to the training points faster than OOD ones. This provides the grounding for why a statistic based on the $k$-NN radius using a label smoothed model is a powerful discriminator. Figure 2 shows the distribution of 1-NN distances for three layers as well as our proposed aggregate statistic on two dataset pairs. Across layers and datasets, we see some separability between in and out-of-distributions points. Label smoothing has the effect of shrinking these distances for both in/out classes but the effect is larger for in points, making the distributions even more separable and thereby improving the performance of our method.

## 3 THEORETICAL RESULTS

In this section, we provide statistical guarantees for using the $k$-NN radius as a method for out of distribution detection. To do this, we assume that the features of the data lie on compact support $\mathcal{X} \subset \mathbb{R}^d$ and that examples are drawn i.i.d. from this. We assume that there exists a density function $f : \mathbb{R}^d \to \mathbb{R}$ corresponding to the distribution of the feature space. This density function can serve as a proxy for how much an example is out of distribution. The difficulty is that this underlying density function is unknown in practice. Fortunately, we can show that the $k$-NN radius method approximates the information conveyed by $f$ based on a finite sample drawn from $f$. For the theory, we define an out of distribution example as an example where $x \notin \mathcal{X}$. Thus, $f(x) = 0$ for such examples.

### 3.1 OUT OF DISTRIBUTION DETECTION HIGH-RECALL RESULT

In this section, we give a result about identifying out of distribution examples based on the $k$-NN radius with perfect recall if we were to use a particular threshold. That is, any example that is indeed out of distribution (i.e. has 0 density) will have $k$-NN radius above that threshold. We also give a guarantee that the false-positives (i.e. those examples with $k$-NN radius higher than that quantity which were not out-of-distribution examples) were of low-density to begin with. Our results hold with high-probability uniformly across all of $\mathbb{R}^d$. As we will see, as $n$ grows and $k/n \to 0$, we find that the $k$-NN radius method using the specified threshold is able to identify which examples are in-distribution vs out-of-distribution.

Our result requires a smoothness assumption on the density function shown below. This smoothness assumption ensures a relationship between the density of a point and the probability mass of balls around that point which is used in the proofs.

**Assumption 1** (Smoothness). *$f$ is $\beta$-Holder continuous for some $0 < \beta \le 1$. i.e. $|f(x) - f(x')| \le C_\beta |x - x'|^\beta$.*

We now give our result below.

**Theorem 1.** *Suppose that Assumption 1 holds and that $0 < \delta < 1$ and $k \ge 2^8 \cdot log(2/\delta)^2 \cdot d \log n$. If we choose*

$$r := \left( \frac{k}{2C_\beta \cdot n \cdot v_d} \right)^{1/(\beta+d)}, \quad \lambda := 5 \cdot C_\beta^{d/(\beta+d)} \cdot \left( \frac{k}{n \cdot v_d} \right)^{\beta/(\beta+d)},$$

*then the following holds uniformly for all $x \in \mathbb{R}^d$ with probability at least $1 - \delta$: (1) If $f(x) = 0$, then $r_k(x) \ge r$. (2) If $r_k(x) \ge r$, then $f(x) \le \lambda$.*

In words, it says that the set of points $x \in \mathbb{R}^d$ satisfying $r_k(x) \gtrsim (k/n)^{1/(\beta+d)}$, is guaranteed to contain all of the outliers and does not contain any points whose density exceeds a cutoff (i.e. $f(x) \gtrsim (k/n)^{\beta/(\beta+d)}$). These quantities all go to 0 as $k/n \to 0$ and thus with enough samples, asymptotically are able to distinguish between out-of-distribution and in-distribution examples.

We can assume the following condition on the boundary smoothness of the density as is done in a recent analysis of $k$-NN density estimation (Zhao & Lai, 2020).

**Assumption 2** (Boundary smoothness). *There exists $0 < \eta \le 1$ such that for any $t > 0$, $f$ satisfies*

$$\mathbb{P}(f(x) \le t) \le C_\eta t^\eta,$$

*where $\mathbb{P}$ represents the distribution of in-distribution examples during evaluation.*

Then, Theorem 1 has the following consequence on the precision and recall of the $k$-NN density based out of distribution detection method.

**Corollary 1.** *Suppose that Assumptions 1 and 2 hold and that $0 < \delta < 1$ and $k \geq 2^8 \cdot log(2/\delta)^2 \cdot d \log n$. Then if we choose*

$$r := \left( \frac{k}{2C_\beta \cdot n \cdot v_d} \right)^{1/(\beta+d)},$$

*then the following holds with probability at least $1 - \delta$. Let us classify an example $x \in \mathbb{R}^d$ as out of distribution if $r_k(x) \geq r$ and in-distribution otherwise. Then, this classifier will identify all of the out-of-distribution examples (perfect recall) and falsely identify in-distribution examples as out-of-distribution with probability (error in precision)*

$$5 \cdot C_\eta \cdot C_\beta^{d/(\beta+d)} \cdot \left( \frac{k}{n \cdot v_d} \right)^{\beta/(\beta+d)}.$$

### 3.2 Ranking preservation result

We next give the following result saying that if the gap in density between two points is large enough, then their rankings will be preserved w.r.t. the $k$-NN radius.

**Theorem 2.** *Suppose that Assumption 1 holds and that $0 < \delta < 1$ and $k \geq 2^8 \cdot log(2/\delta)^2 \cdot d \log n$. Define $C_{\delta,n} := 16 \log(2/\delta)\sqrt{d \log n}$. Then there exists a constant $C$ depending on $f$ such that the following holds with probability at least $1 - \delta$ uniformly for all pairs of points $x_1, x_2 \in \mathbb{R}^d$. If $f(x_1) > f(x_2) + \epsilon_{k,n}$, where*

$$\epsilon_{k,n} := C \left( \frac{C_{\delta,n}}{\sqrt{k}} + (k/n)^{1/d} \right),$$

*then, we have $r_k(x_1) < r_k(x_2)$.*

We note that as $n, k \to \infty$, $k/n \to 0$, and $\log n/\sqrt{k} \to 0$, we have that $\epsilon_{k,n} \to 0$ and thus asymptotically, the $k$-NN radius preserves the ranking by density in the case of non-ties.

### 3.3 Performance under label smoothing embedding hypothesis

In this section, we provide some theoretical intuition behind why the observed label smoothing embedding hypothesis can lead to better performance for the $k$-NN density-based approach on embeddings learned with label smoothing. We make an assumption that our in-distribution has a convex set $\mathcal{X} \subseteq \mathbb{R}^d$ as its support with uniformly lower bounded density and that applying label smoothing has the effect of contracting the space $\mathbb{R}^d$ in the following way: for points in $\mathcal{X}$ the contraction is with respect to a point of origin $x_0$ in the interior of $\mathcal{X}$ so that points in $\mathcal{X}$ move closer to the origin and for outlier points, they move closer to the boundary of $\mathcal{X}$. We ensure that the former happens at a faster rate than the latter and show the following guarantee, which says that under certain regularity conditions on the density and $\mathcal{X}$, we have that the ratio of the $k$-NN distance between an out-of-distribution point and an in-distribution point increases after this mapping. This suggest that under such transformations such as ones induced by what's implied by the label smoothed embedding hypothesis, the $k$-NN distance becomes a better score at separating the in-distribution examples from the out-of-distribution examples.

**Proposition 1** (Improvement of $k$-NN OOD with Label Smoothed Embedding Hypothesis). *Let $f$ has convex and bounded support $\mathcal{X} \subseteq \mathbb{R}^d$ and let $x_0$ be an interior point of $\mathcal{X}$ and additionally assume that there exists $r_0, c_0 > 0$ such that for all $0 < r < r_0$ and $x \in \mathcal{X}$, we have $\text{Vol}(B(x,r) \cap \mathcal{X}) \leq c_0 \cdot \text{Vol}(B(x,r))$ holds (to ensure that $\mathcal{X}$'s boundaries have regularity and are full dimensional) and that $f(x) \geq \lambda_0$ for all $x \in \mathcal{X}$ for some $\lambda_0 > 0$. Define mapping $\phi : \mathbb{R}^d \to \mathbb{R}^d$ such that $\phi(x) = \gamma_{in} \cdot (x - x_0) - x_0$ if $x \in \mathcal{X}$ and otherwise, $\phi(x) = \gamma_{out} \cdot (x - Proj_\mathcal{X}(x)) - Proj_\mathcal{X}(x)$ otherwise, for some $0 < \gamma_{in} < \gamma_{out} < 1$. ($Proj_\mathcal{X}(x)$ denotes the projection of $x$ onto the boundary of convex set $\mathcal{X}$). We see that $\phi$ contracts the points where points in $\mathcal{X}$ contract at a faster rate than those outside of $\mathcal{X}$. Suppose our training set consists of $n$ examples $X_{[n]}$ drawn from $f$, and denote by $\phi(X_{[n]})$ the image of those examples w.r.t. $\phi$.*

*Let $0 < \delta < 1$ and $r_{min} > 0$ and $k$ satisfies*

$$2^8 \cdot \log(2/\delta)^2 \cdot d \log n \leq k \leq \frac{1}{2} c_0 \cdot v_d \cdot \left( \frac{\gamma_{out} - \gamma_{in}}{\gamma_{in}} \cdot r_{min} \right)^d \cdot n$$

*and $n$ is sufficiently large depending on $f$. Then with probability at least $1 - \delta$, the following holds uniformly among all $r_{min} > 0$, choices of $x_{in} \in \mathcal{X}$ (in-distribution example) and $x_{out}$ such that $d(x, \mathcal{X}) \geq r_{min}$ (out-of-distribution example with margin). The following holds.*

$$\frac{r_k(\phi(x_{out}); \phi(X_{[n]}))}{r_k(\phi(x_{in}); \phi(X_{[n]}))} > \frac{r_k(x_{out}; X_{[n]})}{r_k(x_{in}; X_{[n]})},$$

*where $r_k(x, A)$ denotes the $k$-NN distance of $x$ w.r.t. dataset $A$.*

## 4 EXPERIMENTS

### 4.1 SETUP

We validate our method on MNIST (LeCun et al., 1998), Fashion MNIST (Xiao et al., 2017), SVHN (cropped to 32x32x3) (Netzer et al., 2011), CIFAR10 (32x32x3) (Krizhevsky et al., 2009), and CelebA (32x32x3) (Liu et al., 2015). In CelebA, we train against the binary label "smiling". We train models on the train split of each of these datasets, and then test OOD binary classification performance for a variety of OOD datasets, while always keeping the in-distribution to be the test split of the dataset used for training. Thus, a dataset pairing denoted "A $\rightarrow$ B" means that the classification model is trained on A's train and is evaluated for OOD detection using A's test as in-distribution points and B's test as out-of-distribution points. In addition to the aforementioned, we form OOD datasets by corrupting the in-distribution test sets - by flipping images left and right (HFlip) as well as up and down (VFlip) - and we also use the validation split of ImageNet (32x32x3), the test splits of KMNIST (28x28x1), EMNIST digits (28x28x1), and Omniglot (32x32x3). All datasets are available as Tensorflow Datasets [1]. We measure the OOD detectors' ROC-AUC, sample-weighting to ensure balance between in and out-of-distribution samples (since they can have different sizes). For MNIST and Fashion MNIST, we train a 3-layer ReLU-activated DNN, with 256 units per layer, for 20 epochs. For SVHN, CIFAR10, and CelebA, we train the convolutional LeNet5 (LeCun et al., 2015) for 10 epochs. We use 128 batch size and Adam optimizer with default learning rate 0.001 throughout. All methods were implemented in TensorFlow and trained on a cloud environment. We estimate we used a total of 10k CPU hours for all of our experiments. For embedding-based methods, we aggregate over 3 layers for the DNN and 4 dense layers for LeNet5, including the logits. For our method, we always use Euclidean distance between embeddings, $k = 1$ and label smoothing $\alpha = 0.1$. These could likely be tuned for better performance in the presence of a validation OOD dataset sufficiently similar to the unknown test set. We do not do this since we assume the absence of such dataset.

### 4.2 BASELINES

We validate our method against the following recent baselines:
**Control.** We use the model's maximum softmax confidence, as suggested by Hendrycks & Gimpel (2016). The lower the confidence, the more likely the example is to be OOD.
**Robust Deep $k$-NN.** This method, proposed in Papernot & McDaniel (2018) leverages $k$-NN for a query input as follows: it computes the label distribution of the query point's nearest training points for each layer and then computes a layer-aggregated $p$-value-based non-conformity score against a held-out calibration set. Queries that have high disagreement, or impurity, in their nearest neighbor label set are suspected to be OOD. We use 10% of the training set for calibration, $k = 50$, and cosine similarity, as described in the paper.
**DeConf.** Hsu et al. (2020) improves over the popular method ODIN (Liang et al., 2017) by freeing it from the needs of tuning on OOD data. It consists of two components - a learned "confidence decomposition" derived from the model's penultimate layer, and a modified method for perturbing inputs optimally for OOD detection using a Fast-Sign-Gradient-esque strategy. We use the "h" branch of the cosine similarity variant described in the paper. We searched the perturbation hyperparameter $\epsilon$

---

[1] https://www.tensorflow.org/datasets

| Dataset | Control | $k$-NN (0.1 LS) | $k$-NN (no LS) | DeConf | Robust $k$-NN | SVM | Isolation Forest |
|---|---|---|---|---|---|---|---|
| | | | Train/In: *MNIST* | | | | |
| EMNIST | 0.835 | 0.950 | **0.966** | 0.693 | 0.875 | 0.794 | 0.346 |
| F. MNIST | 0.838 | **0.968** | 0.954 | 0.747 | 0.904 | 0.569 | 0.655 |
| KMNIST | 0.882 | **0.984** | **0.985** | 0.746 | 0.923 | 0.756 | 0.358 |
| HFlip | 0.852 | **0.914** | 0.871 | 0.706 | 0.847 | 0.568 | 0.559 |
| VFlip | 0.833 | **0.883** | 0.840 | 0.684 | 0.812 | 0.537 | 0.599 |
| | | | Train/In: *Fashion MNIST* | | | | |
| EMNIST | 0.551 | **0.993** | 0.983 | 0.670 | 0.756 | 0.881 | 0.170 |
| HFlip | 0.557 | **0.730** | 0.698 | 0.581 | 0.608 | 0.616 | 0.443 |
| VFlip | 0.642 | **0.915** | 0.875 | 0.704 | 0.774 | 0.700 | 0.442 |
| KMNIST | 0.673 | **0.989** | 0.962 | 0.759 | 0.814 | 0.818 | 0.268 |
| MNIST | 0.697 | **0.997** | 0.969 | 0.837 | 0.854 | 0.782 | 0.338 |
| | | | Train/In: *SVHN* | | | | |
| CelebA | 0.785 | **0.906** | 0.857 | 0.682 | 0.887 | 0.702 | 0.261 |
| CIFAR10 | 0.821 | 0.855 | 0.722 | 0.693 | **0.873** | 0.564 | 0.423 |
| CIFAR100 | 0.820 | **0.876** | 0.755 | 0.682 | **0.878** | 0.585 | 0.385 |
| ImageNet | 0.825 | 0.852 | 0.723 | 0.693 | **0.876** | 0.560 | 0.416 |
| Omniglot | 0.685 | **0.977** | 0.958 | 0.521 | 0.861 | 0.884 | 0.093 |
| HFlip | 0.737 | 0.683 | 0.580 | 0.667 | **0.746** | 0.504 | 0.554 |
| VFlip | 0.674 | 0.648 | 0.573 | 0.604 | **0.686** | 0.515 | 0.533 |
| | | | Train/In: *CIFAR10* | | | | |
| CelebA | 0.570 | **0.780** | **0.764** | 0.521 | 0.637 | 0.657 | 0.387 |
| CIFAR100 | **0.633** | 0.598 | 0.573 | 0.584 | 0.615 | 0.485 | 0.514 |
| HFlip | 0.503 | **0.512** | **0.513** | 0.502 | **0.512** | 0.500 | 0.502 |
| VFlip | **0.645** | 0.594 | 0.580 | 0.583 | 0.616 | 0.471 | 0.531 |
| ImageNet | **0.639** | 0.588 | 0.562 | 0.583 | 0.620 | 0.448 | 0.562 |
| Omniglot | 0.356 | **0.960** | **0.980** | 0.462 | 0.587 | 0.954 | 0.064 |
| SVHN | **0.725** | 0.381 | 0.384 | 0.584 | 0.635 | 0.323 | 0.677 |
| | | | Train/In: *CelebA* | | | | |
| HFlip | 0.501 | **0.504** | **0.503** | 0.500 | 0.501 | 0.501 | 0.498 |
| VFlip | 0.459 | **0.738** | **0.696** | 0.354 | 0.481 | 0.610 | 0.277 |
| CIFAR100 | 0.639 | **0.689** | 0.607 | 0.535 | 0.652 | 0.418 | 0.426 |
| CIFAR10 | 0.638 | **0.692** | 0.605 | 0.529 | 0.642 | 0.422 | 0.431 |
| ImageNet | 0.647 | **0.684** | 0.598 | 0.535 | 0.648 | 0.412 | 0.436 |
| Omniglot | 0.586 | **0.910** | **0.899** | 0.480 | 0.654 | 0.573 | 0.079 |
| SVHN | **0.612** | 0.520 | 0.441 | 0.539 | **0.586** | 0.411 | 0.546 |

Table 1: ROC-AUC for different methods and dataset pairings. The datasets enclosed by double lines represent the training and in-distribution test set, while the datasets listed beneath them are used as OOD. Each entry was run 5 times. The standard errors are quite small, with a median of 0.0071. Entries within two standard errors of the max are bolded. We see that label smoothing almost always improves the performance of our method and that the method is competitive across a variety of datasets.

over the range listed in the paper, but found that it never helped OOD in our setting. We thus reports numbers for $\epsilon = 0$.

**SVM.** We learn a one-class SVM (Schölkopf et al., 1999) on the intermediate embedding layers and then aggregate the outlier scores across layers in the same way we propose in our method. We use an RBF kernel.

**Isolation Forest.** This is similar to SVM, but uses an isolation forest (Liu et al., 2008) with 100 estimators at each layer.

## 4.3 RESULTS

Our main results are shown in Table 1. We observe that label smoothing nearly always improved our method, denoted $k$-NN, and that the method is competitive, outperforming the rest on the most

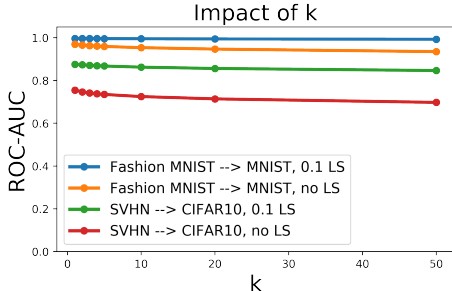 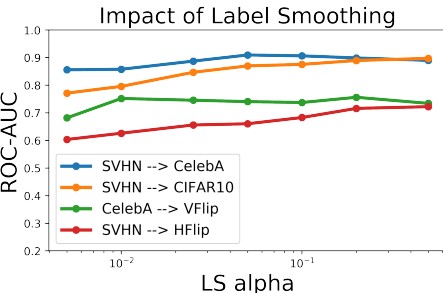

Figure 2: **Left:** Impact of $k$ on ROC-AUC. We observe that performance is mostly stable across a range of $k$. We do see slight degradation with larger $k$, and so we recommend users a default of $k = 1$. **Right:** Impact of $\alpha$ on ROC-AUC for four dataset pairings. Note that the x-axis is log-scale and the y-axis is zoomed in. We generally see that performance improves with larger $\alpha$ until it reaches a critical point, after which it declines. While this critical point is model and data dependent, we see that blithely selecting a fixed value like 0.1 results in reasonable performance.

| | | Depth (from logits) | | |
|---|---|---|---|---|
| OOD | LS | 0 | -1 | -2 |
| Train/In: FashionMnist | | | | |
| EMNIST | 0.0 | 0.970 | **0.984** | 0.976 |
| | 0.1 | 0.973 | **0.993** | **0.994** |
| KMNIST | 0.0 | 0.927 | 0.963 | **0.970** |
| | 0.1 | 0.973 | 0.986 | **0.988** |
| MNIST | 0.0 | **0.958** | **0.966** | 0.957 |
| | 0.1 | 0.992 | **0.996** | 0.992 |

| | | Depth (from logits) | | | |
|---|---|---|---|---|---|
| OOD | LS | 0 | -1 | -2 | -3 |
| Train/In: CelebA | | | | | |
| CIFAR100 | 0.0 | 0.472 | **0.722** | 0.638 | 0.583 |
| | 0.1 | 0.565 | **0.755** | 0.692 | 0.560 |
| CIFAR10 | 0.0 | 0.474 | **0.717** | 0.641 | 0.606 |
| | 0.1 | 0.566 | **0.760** | 0.699 | 0.583 |
| ImageNet | 0.0 | 0.475 | **0.709** | 0.622 | 0.583 |
| | 0.1 | 0.566 | **0.750** | 0.682 | 0.566 |
| Omniglot | 0.0 | 0.586 | 0.959 | 0.964 | **0.993** |
| | 0.1 | 0.658 | 0.956 | 0.959 | **0.990** |

Figure 3: We observe the ROC-AUC of our method using only a single layer at a time, for Fashion MNIST and CelebA, with and without label smoothing. We find that label smoothing usually helps every layer on its own, and that the penultimate layer (depth = -1) often outperforms the rest on these datasets.

number of dataset pairs. SVM, Isolation Forest serve as key ablative models, since they leverage the same intermediate layer representations as our method and their layer-level scores are combined in the same way. Interestingly, we see that the $k$-NN consistently outperforms them, revealing the discriminative power of the $k$-NN radius distance. Robust $k$-NN also uses the same layer embeddings and $k$-NN, but in a different manner. Crucially, it performs OOD detection by means of the nearest training example neighbors' class label distribution. Given that we outperform Robust $k$-NN more often than not, we might conjecture that the *distance* has more discriminative power for OOD detection than *class label distribution*. We were surprised that DeConf routinely did worse than the simple control, despite having implementing the method following the paper closely.

## 4.4 ABLATIONS

In this section, we study the impact of three factors on our method's performance: (1) the number of neighbors, $k$, (2) the amount of label smoothing $\alpha$, and (3) the intermediate layers used.

**Impact of $k$.** In Figure 2 we plot the impact of $k$ on OOD detection for two dataset pairings: MNIST $\rightarrow$ Fashion MNIST and SVHN $\rightarrow$ CIFAR10 with and without label smoothing. We see that larger $k$ degrades ROC-AUC monotonically, but the effect is rather small. We thus recommend a default of $k = 1$. $k = 1$ has the added benefit of being more efficient in most implementations of index-based large-scale nearest-neighbor lookup systems.

**Impact of Label Smoothing $\alpha$.** We now consider the effect of label smoothing amount $\alpha$ on ROC-AUC in Figure 2. We see, interestingly, that performance mostly increases monotonically with larger

$\alpha$ until it reaches a critical point, after which it declines monotonically. While this optimal point may be data and model dependent and thus hard to estimate, we've found that selecting a fixed value like 0.1 works well in most cases.

**Impact of Intermediate Layer**. Our method aggregates $k$-NN distance scores across intermediate layers. We depict the effect of different choices of a *single* layer on Fashion MNIST and CelebA in Figure 3. We find that label smoothing generally boosts performance for each layer individually and that while no single layer is always optimal, the penultimate layer performs fairly well across the datasets.

## 5 RELATED WORK

**Out-of-Distribution Detection.** OOD detection has classically been studied under names such as outlier, anomaly, or novelty detection. One line of work are density-based methods: Ester et al. (1996) presents a density-based clustering algorithm which is also an outlier detection algorithm by identifying *noise* points which are points whose $\epsilon$-neighborhood has fewer than a certain number of points. Breunig et al. (2000); Kriegel et al. (2009) propose local outlier scores based on the degree to which how isolated the datapoint is with respect to its neighborhood via density estimation. Another line of work uses $k$-NN density estimates (Ramaswamy et al., 2000; Angiulli & Pizzuti, 2002; Hautamaki et al., 2004; Dang et al., 2015). We use the $k$-NN density estimator, but use it in conjunction with the embeddings of a neural network trained with label smoothing. Other classical approaches include the one-class SVM (Schölkopf et al., 2001; Chen et al., 2001), isolation forest (Liu et al., 2008). A slew of recent methods have been proposed for OOD. We refer interested readers to a survey.

**Label Smoothing.** Label smoothing has received much attention lately; we give a brief review here. It has been shown to improve model calibration (and therefore the generation quality of auto-regressive sequence models like machine translation) but has been seen to hurt teacher-to-student knowledge distillation (Pereyra et al., 2017; Xie et al., 2016b; Chorowski & Jaitly, 2016; Gao et al., 2020; Lukasik et al., 2020b; Müller et al., 2019). Müller et al. (2019) show visually that label smoothing encourages the penultimate layer representations of the training examples from the same class to group in tight clusters. Lukasik et al. (2020a) shows that label smoothing makes models more robust to label noise in the training data (to a level competitive with noisy label correction methods), and, furthermore, smoothing the teacher is beneficial when distilling from noisy data. Chen et al. (2020) corroborates the benefits of smoothing for noisy labels and provides a theoretical framework wherein the optimal smoothing parameter $\alpha$ can be identified. LS has been seen to hurt performance on sparse distributions (Meister et al., 2020) and decrease robustness to adversarial attacks (Zantedeschi et al., 2017). Yuan et al. (2020) casts knowledge distillation (KD) as a type of learned label smoothing regularization, showing that part of KD's success stems from its ability to regularize soft labels in the same way as LS. They then propose Teacher-free KD that achieves comparable performance to normal KD with a superior teacher.

**$k$-NN Density Estimation Theory.** Statistical guarantees for $k$-NN density estimation has had a long history (Fukunaga & Hostetler, 1973; Devroye & Wagner, 1977; Mack, 1983; Buturović, 1993; Biau et al., 2011; Kung et al., 2012). Most works focus on showing convergence guarantees under metrics like $L_2$ risk or are asymptotic. Dasgupta & Kpotufe (2014) provided the first finite-sample *uniform* rates, which to our knowledge is the strongest result so far. Our analysis uses similar techniques, which they also borrow from Chaudhuri & Dasgupta (2010); however our results are for the application of OOD detection whereas Dasgupta & Kpotufe (2014)'s goal was mode estimation. As a result, our results hold with high probability uniformly in the input space, while having finite-sample guarantees and provide new theoretical insights into the use of $k$-NN for OOD detection.

## 6 CONCLUSION

In this work we put forward the Label Smoothing Embedding Hypothesis and proposed a deep $k$-NN density-based method for out-of-distribution detection that leverages the separability of intermediate layer embeddings and showed how label smoothing the model improves our method.

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

# Appendix

## A PROOFS

We need the following result giving guarantees between the probability measure on the true balls and the empirical balls.

**Lemma 1** (Uniform convergence of balls (Chaudhuri & Dasgupta, 2010)). *Let $\mathcal{F}$ be the distribution corresponding to $f$ and $\mathcal{F}_n$ be the empirical distribution corresponding to the sample $X$. Pick $0 < \delta < 1$. Assume that $k \geq d \log n$. Then with probability at least $1 - \delta$, for every ball $B \subset \mathbb{R}^D$ we have*

$$\mathcal{F}(B) \geq C_{\delta,n} \frac{\sqrt{d \log n}}{n} \Rightarrow \mathcal{F}_n(B) > 0$$

$$\mathcal{F}(B) \geq \frac{k}{n} + C_{\delta,n} \frac{\sqrt{k}}{n} \Rightarrow \mathcal{F}_n(B) \geq \frac{k}{n}$$

$$\mathcal{F}(B) \leq \frac{k}{n} - C_{\delta,n} \frac{\sqrt{k}}{n} \Rightarrow \mathcal{F}_n(B) < \frac{k}{n},$$

*where $C_{\delta,n} = 16 \log(2/\delta) \sqrt{d \log n}$*

*Remark.* For the rest of the paper, many results are qualified to hold with probability at least $1 - \delta$. This is precisely the event in which Lemma 1 holds.

*Remark.* If $\delta = 1/n$, then $C_{\delta,n} = O((\log n)^{3/2})$.

*Proof of Theorem 1.* Suppose that $x$ satisfies $f(x) = 0$. Then we have

$$\mathcal{F}(B(x,r)) = \int f(x') \cdot 1[x' \in B(x,r)]dx' = \int |f(x') - f(x)| \cdot 1[x' \in B(x,r)]dx'$$

$$\leq C_\beta \int |x' - x|^\beta \cdot 1[x' \in B(x,r)]dx' \leq C_\beta r^{\beta+d} v_d = \frac{k}{2n} \leq \frac{k}{n} - C_{\delta,n} \frac{\sqrt{k}}{n}.$$

Therefore, by Lemma 1, we have that $r_k(x) \leq r$. Now for the second part, we prove the contrapositive. Suppose that $f(x) > \lambda$. Then we have

$$\mathcal{F}(B(x,r)) \geq (\lambda - C_\beta r^\beta) \cdot v_d \cdot r^d \geq \frac{2k}{n} \geq \frac{k}{n} + C_{\delta,n} \frac{\sqrt{k}}{n}.$$

Therefore, by Lemma 1, we have that $f(x) \geq \lambda$, as desired. $\square$

*Proof of Theorem 2.* We borrow some proof techniques used in Dasgupta & Kpotufe (2014) to give uniform bounds on $|f(x) - f_k(x)|$ where $f_k$ is the $k$-NN density estimator defined as

$$f_k(x) := \frac{k}{n \cdot v_d \cdot r_k(x)^d}.$$

It is also clear that $r \leq C' \cdot (k/n)^{1/d}$ for some $C'$ depending on $f$. If we choose $r$ such that

$$\mathcal{F}(B(x,r)) \leq v_d r^d (f(x) + C_\beta r^\beta) = \frac{k}{n} - C_{\delta,n} \frac{\sqrt{k}}{n}.$$

Then, we have by Lemma 1 that $r_k(x) > r$. Thus, we have

$$f_k(x) < \frac{k}{n \cdot v_d \cdot r^d} = \frac{f(x) + C_\beta r^\beta}{1 - C_{\delta,n}/\sqrt{k}} \leq f(x) + C_1 \cdot \left( \frac{C_{\delta,n}}{\sqrt{k}} + (k/n)^{1/d} \right)$$

for some $C_1 > 0$ depending on $f$. The argument for the other direction is similar: we instead choose $r$ such that

$$\mathcal{F}(B(x,r)) \geq v_d r^d (f(x) - C_\beta r^\beta) = \frac{k}{n} + C_{\delta,n} \frac{\sqrt{k}}{n}.$$

Again it's clear that $r \leq C'' \cdot (k/n)^{1/d}$ for some $C''$ depending on $f$. Next, we have by Lemma 1 that $r_k(x) \leq r$. Thus, we have

$$f_k(x) \geq \frac{k}{n \cdot v_d \cdot r^d} = \frac{f(x) - C_\beta r^\beta}{1 + C_{\delta,n}/\sqrt{k}} \geq f(x) - C_2 \cdot \left( \frac{C_{\delta,n}}{\sqrt{k}} + (k/n)^{1/d} \right)$$

for some $C_2$ depending on $f$. Therefore, there exists $C_0$ depending on $f$ such that

$$\sup_{x \in \mathbb{R}^d} |f(x) - f_k(x)| \leq C_0 \left( \frac{C_{\delta,n}}{\sqrt{k}} + (k/n)^{1/d} \right).$$

Finally, we have that setting $C = 2 \cdot C_0$, we have

$$f_k(x_1) \geq f(x_1) - C_0 \cdot \left( \frac{C_{\delta,n}}{\sqrt{k}} + (k/n)^{1/d} \right) > f(x_2) + C_0 \cdot \left( \frac{C_{\delta,n}}{\sqrt{k}} + (k/n)^{1/d} \right) \geq f_k(x_2),$$

then it immediately follows that $r_k(x_1) < r_k(x_2)$, as desired. $\qquad \square$

*Proof of Proposition 1.* Define $r_u := \max_{x \in \mathcal{X}} r_k(x)$. We have

$$\frac{r_k(\phi(x_{out}); \phi(X_{[n]}))}{r_k((x_{out}; X_{[n]}))} \geq \frac{d(\phi(x_{out}), \phi(X))}{d(x_{out}, X) + r_u} = \frac{\gamma_{out} \cdot d(x_{out}, X)}{d(x_{out}, X) + r_u}.$$

Next, we have

$$\frac{r_k(\phi(x_{in}); \phi(X_{[n]}))}{r_k(x_{in}; X_{[n]})} = \gamma_{in},$$

since all pairwise distances within $\mathcal{X}$ are scaled by $\gamma_{in}$ through our mapping $\phi$.

Thus, it suffices to have

$$\gamma_{in} \leq \frac{\gamma_{out} d(x_{out}, X)}{d(x_{out}, X) + r_u},$$

which is equivalent to having

$$r_u \leq \frac{\gamma_{out} - \gamma_{in}}{\gamma_{in}} d(x, \mathcal{X}).$$

Which holds when

$$r_u \leq \frac{\gamma_{out} - \gamma_{in}}{\gamma_{in}} \cdot r_{min}.$$

This holds because we have $r_u \leq \left( \frac{2k}{c_0 n v_d} \right)^{1/d}$ by Lemma 1, and the result follows by the condition on $k$. $\qquad \square$

