# OpenReview forum: "Label Smoothed Embedding Hypothesis for Out-of-Distribution Detection"
_ICLR.cc/2022/Conference — ICLR 2022 Submitted_

### Official Review · Reviewer_swL5 · 2021-11-02

**Correctness:** 3
**Technical Novelty And Significance:** 3
**Empirical Novelty And Significance:** 3
**Recommendation:** 5
**Confidence:** 3

**Main Review:**

# Strengths

This paper proposed an interesting unsupervised OOD method. Empirical experiments showed the superior performance of the proposed method over some baselines; meanwhile, theoretical high-probability statistical results were given to analyze the proposed method.

# Weaknesses

Here are some of my comments/questions. I would appreciate it if the author(s) could give a response. If I am wrong, please correct me. Thanks.

- The author(s) demonstrated the effectiveness of the proposed method through empirical experiments. Currently, there are many new developments in OOD (some of which can be quickly searched and found through search engines), but the methods compared in this paper are obviously inadequate. So I encourage the author(s) might provide more up-to-date OOD algorithms as baseline comparisons before this paper is published at influential conferences or journals so that the effectiveness of the proposed method is convincing.

In addition, this paper is generally well written, but some places (some issues) in this paper should be further clarified/fixed.

- The format of many references is inconsistent, for example,

   - Conference names are sometimes abbreviated and sometimes not;

   - Authors' names are sometimes abbreviated and sometimes not;

- Some tables are presented as a Figure, resulting in inconsistent font sizes in the tables; for example, Figure 3.


**Summary Of The Paper:**

Out-of-distribution detection is important in many aspects of machine learning. Based on label smoothing and $k$-NN density estimate, this paper proposed an unsupervised OOD detection method. By empirical experiments, the author(s) demonstrated the effectiveness of the proposed method. And theoretical results for the proposed OOD method were provided.

**Summary Of The Review:**

This paper proposed a deep $k$-NN density-based method for out-of-distribution detection in an unsupervised fashion. Empirical experiments were performed to verify the effectiveness of the proposed method, and theoretical results were proven. However, I think that while this is an interesting paper, it would be better to compare more of the new OOD baselines to make the proposed method persuasive.

---

### Official Review · Reviewer_orsU · 2021-11-05

**Correctness:** 4
**Technical Novelty And Significance:** 3
**Empirical Novelty And Significance:** 2
**Recommendation:** 6
**Confidence:** 4

**Main Review:**

1. Strengths.

- The paper is well written and easy to follow.

- It proposes a simple but effective method for OOD detection based on the k-nn density estimate in the hidden space of a classifier.

- The paper provides interesting theoretical results regarding the importance of the k-nn radius for OOD prediction.

- The experimental results reflect the benefits of label smoothing on the OOD prediction task. Moreover, the proposed method offers competitive performance compared to the chosen baselines.

2. Points that can be improved.

- Most of the provided theoretical results seem to be on the properties of the k-nn radius for OOD detection. However, it would make more sense to derive similar results for the statistic T(x), which is used as the OOD score.

- The paper includes some interesting results regarding the recall/prediction of the k-NN density-based OOD detection method. It would be useful to report these measures in the experiments (e.g., the area under the precision-recall curve).

- The different theoretical results assume that the number of nearest neighbors is relatively large. Based on the experiments, however, the authors suggest setting k = 1. More discussions regarding this aspect would be useful.

- The choice of the threshold "t" to make the OOD decision does not seem to be discussed in the paper nor investigated in the experiments.

3. Question

One question (related to the first point of section 2. above) that arises is why not make OOD prediction based on the k-nn radius rk(gi(x); gi(Xin)) directly.



**Summary Of The Paper:**

This paper proposes a method for Out-Of-Distribution (OOD) detection that relies on the k-nn density estimate in the feature space of a classifier model. Given a data example, the idea is to compute an OOD score based on the k-nn radius w.r.t to the training observations at every hidden layer. The mean of the scores from the different layers is then used for OOD prediction. The authors provide several theoretical results highlighting some desirable properties of the k-nn radius for the OOD detection task. One such result states that training the classifier with label smoothing can improve k-nn radius-based OOD prediction under some regularity conditions. Empirical results show that the proposed method performs well compared to the chosen baselines, and label smoothing seems to improve OOD predictions in most cases. Experiments also include ablation studies showing that the proposed method is not very sensitive to some hyperparameter settings, such as the number of nearest neighbors and the amount of smoothing.

**Summary Of The Review:**

The paper is well-written. It proposes a simple method for OOD detections along with interesting theoretical analyses and promising empirical results. There are, however, several aspects that deserve to be improved/clarified. Please refer to the main review section for more details.

---

### Official Review · Reviewer_pUXL · 2021-11-05

**Correctness:** 3
**Technical Novelty And Significance:** 2
**Empirical Novelty And Significance:** 2
**Recommendation:** 5
**Confidence:** 3

**Main Review:**


Pros
- The paper is mostly well-written and easy to follow.
- Although it is not connected to the actual implementation, the theoretical results that guarantee there exists a threshold that perfectly separate OOD and in-distribution samples and show that label smoothing can improve the OOD detection performance are interesting.

Cons
- Computation of r_k(x;X) for each new sample x involves computation of distance between the feature of the sample and features of all existing training dataset. Due to this huge computation (or memory requirement in the case of storing all feature values of training samples), the method will not be applicable to modern large-scale datasets.
- Also, it is unclear how the authors obtain the threshold between in-distribution samples and OOD samples in terms of r_k(x;X) values.
In deep architectures, a range of feature values is largely different from layer to layer. Therefore, it seems unnatural to use the average of \hat{T}_i(x) as a threshold. It would be better to investigate a more principled way to weigh \hat{T}_i(x) or at least include how \hat{T}_i(x) differs across the different layers.
- The theoretical results do not go well with the practice. Specifically, the theoretical results in the paper are based on the conditions that k \geq f(log n) for some function f( ) and k/n \rightarrow \infty. This means that it assumes a large scale dataset, and the value of k will be large as well. However, the proposed method is not applicable to the large dataset and the authors performed experiments with k=1.
Experiments are performed only with small scale dataset and a shallow architecture. Therefore, it is unclear whether the method still outperforms baselines in the popular benchmark or in practice.

**Summary Of The Paper:**

This paper proposes a OOD detection method based on the k-nn density estimator. The authors provide theoretical results that guarantee the existence of a threshold separating in-distribution and OOD samples in terms of k-nn radius. In addition, the authors propose to use their method along with label smoothing based on the empirical observation in Muller 2020 that shows label smoothing helps to gather in-class samples and separate samples from different classes in feature space. They also provide a theoretical result that explains how label smoothing can help to improve the OOD detection performance of the method.


**Summary Of The Review:**

While the theoretical results in the paper are interesting, these results are not connected to the implementation of the methods. Also, the method has a limited practical significance in terms of scalability.

---

### Official Review · Reviewer_E77y · 2021-11-08

**Correctness:** 2
**Technical Novelty And Significance:** 2
**Empirical Novelty And Significance:** 3
**Recommendation:** 3
**Confidence:** 4

**Main Review:**

While from the experimental results it appears that label smoothing generally improves kNN based OOD detection, I have many issues with the paper:

1. The paper extrapolates the conclusions of Muller et al too much. In Muller et al, the authors showed that label smoothing pushes the penultimate layer representation of a class equally away from the representations of other classes. This conclusion was  grounded in simple but concrete analytical results as well as compelling empirical demonstrations. This in no way implies that label smoothing pushes "OOD points further away from the training manifold". Note that Muller's conclusions are only applicable to the classification layer representations and not representations learned by deeper layers.  Furthermore, uniform label smoothing doesn't respect relationships between class labels and this clustering might hurt OOD detection for some domains. I find designing algorithms for high-dimensional representations based on intuition gained from 2D visualizations to be deeply problematic.
2. The claim in abstract that with label smoothing "kNN density estimator performs better as a OOD detection method theoretically" is not supported at all. In Proposition 1 the authors assume that label smoothing separates the in and out-distribution points, i.e. the representations of ID points are pushed towards a single point in the interior of the manifold while the representation of OOD points are pushed towards the boundary. Under this assumption it is obvious that KNN would work well for OOD detection but the theoretical result is the assumption.
3. The **Label Smoothed Embedding Hypothesis** formalism in in Proposition 1 is too strong and doesn't make sense. In Proposition 1 the authors assume that there is a single point $x_0$ in the interior of the feature space $\mathcal{X}$ towards which all the in distribution points are pushed towards when using label smoothing. Also, I don't understand how the feature representation of $x$ ($\phi(x)$) is a linear combination of $x$ and $x_0$.
4. The writing is too informal. In Definition 1, $B(x, r)$ is not defined which I assume is the Euclidean ball of radius r centered on x. Terms like training manifold are not defined. It is not clear what "uncertainty in the softmax function" means. The authors claim that softmax probabilities cannot distinguish between ID and and OOD points because "softmax probabilities sum to 1". I don't understand what this means. Any discrete distribution must sum to 1.
5. Figure 1 doesn't support claim that "1NN radii shrink, indicating a contraction towards training manifold for both in-distribution and OOD points. Contraction is higher for ID points than OOD points.". In the top-right corner even though the 1NN radii shrink under label smoothing, it seems like there is high overlap between the 1NN radii of ID and OOD points. There has to be some quantitative justification for this claim, it is hard to conclude this from looking at the plots.
6. Following up from the claim about not using softmax distribution, I would have liked to see the control implemented with temperature scaling to reduce calibration error first. Using the softmax probabilities naively to detect OOD points is not a good idea since it is well known that neural network probabilities are not well calibrated. Simple temperature scaling can fix this.

**Summary Of The Paper:**

The paper proposes using kNN density estimation for detecting outliers or out-of-distribution (OOD) points using hidden representations learned by a neural network trained with label smoothing. The main claim of the paper is that, under the hypothesis that "label smoothing causes in-distribution points to contract to the training points faster than OOD points", the kNN radius can be used to separate OOD points. The paper provides some weak empirical justification for this hypothesis but demonstrate that their method generally out-performs kNN without label smoothing and other baselines on 5 datasets.

**Summary Of The Review:**

While I think there is some merit in the proposed algorithm, a lot of the claims made in the paper are not well supported and are extrapolated too much from the results of Muller et al. The theoretical results are perfunctory and the writing it very informal. I think the paper can be improved by making the claims more rigorous either empirically or analytically, especially the label smoothing embedding hypothesis.

---

### Decision · Program_Chairs · 2022-01-20

**Decision:**

Reject

**Comment:**

The problem considered in this paper is of general interest to all reviewers. However, while the reviewers in general appreciate the authors’ effort in providing theoretical analysis for a seemingly effective algorithm, they are unconvinced that the key technical claims are well justified (i.e. separation between theoretical analysis and the algorithm, which ultimately relies on the OOD score), the propositions are clear (e.g., key claims in the quality of kNN density estimator as an OOD detector not well supported by analysis/experiments), or that the experimental results are sufficiently compelling (e.g., lack of controlled experiments/ ablation study) to merit acceptance for the proposed solution.